# The Current Health and Wellbeing of the Survivors of the Rana Plaza Building Collapse in Bangladesh: A Qualitative Study

**DOI:** 10.3390/ijerph16132342

**Published:** 2019-07-02

**Authors:** Humayun Kabir, Myfanwy Maple, Md Shahidul Islam, Kim Usher

**Affiliations:** 1School of Health, University of New England, Armidale, NSW 2351, Australia; 2Department of Sociology, University of Dhaka, Dhaka-1000, Bangladesh

**Keywords:** disaster, workplace safety, workplace environment, physical and psychological health vulnerabilities, readymade garment worker, qualitative research, Bangladesh

## Abstract

This study aims to identify the ongoing physical and psychological health vulnerabilities of the readymade garment (RMG) factory workers involved in the Rana Plaza building collapse in 2013, along with their experiences within the current socioeconomic and political contexts of Bangladesh. Seventeen Rana Plaza survivors participated in unstructured, face-to-face, in-depth interviews. Interviews were thematically analyzed using Haddon’s matrix to examine pre-event, event, and post-event injury experiences. The collapse of the Rana Plaza building resulted in significant physical and emotional trauma for those who survived the event. The majority of the participants were forced to attend work on the day of the collapse. Participants reported physical health complaints related to bone injuries/fractures and amputation, severe headache, kidney problems, and functional difficulties. In addition to the reported physical health issues, the participants revealed psychological health issues including trauma, depression and suicidal ideation, sleep disorders, anxiety, and sudden anger. Participants described barriers to their potential for re-employment in the RMG sector and outlined their limited access to free healthcare for follow-up treatment. Those who survived the collapse of the Rana Plaza building continue to experience significant adverse physical and emotional outcomes related to the disaster. Yet, they have little recourse to ensure the availability of adequate health care and rehabilitation. Given the international reliance on the Bangladeshi RMG industry, continued pressure to ensure care is provided for these survivors, and to reduce the risk of future disasters, is necessary.

## 1. Introduction

The collapse of the Rana Plaza in Savar Upazila, Dhaka, Bangladesh on 24 April 2013, was the worst and deadliest garment factory disaster on record [1,2]. More than 1100 workers were killed, approximately 2500 injured, and about 100 left missing among the 5000 workers in the various factories located at the Rana Plaza on that day [3,4,5,6,7,8,9,10,11]. The collapse of the eight-story building, which housed a private bank, apartments, stores, and clothing factories, occurred the day after the discovery of a crack on the third floor [1]. While the bank and some of the other shops closed immediately when the crack was noticed [7,12], the management of the clothing factories continued production and remained open until the time of the collapse. 

Incidents like the Rana Plaza collapse are not new in the Bangladesh readymade garment (RMG) sector. Previously, 64 workers died in 2005 in a factory building collapse in the same area [1]. In addition, in 2012, a fire in the Tazreen Fashion factory, Ashulia (approximately 24 km from Rana Plaza) killed about 112 workers, and left more than 53 workers with serious burn injuries [6,13]. Since the Rana Plaza collapse, there have been further disasters of a similar nature in the RMG industry, including a fire in the Multifabs limited factory located at Gazipur (approximately 29.2 km from Rana Plaza site) in 2017 [14], signifying that disasters in the Bangladesh RMG sector remain an ongoing concern. According to the International Labor Rights Forum Research (ILRF), more than 700 garment workers (excluding the Rana Plaza collapse) have died as a result of unsafe buildings in Bangladesh since 2005 [15]. However, the Rana Plaza disaster remains the most significant disaster in the history of Bangladesh given the scale of the collapse and resulting deaths and injuries it caused.

The collapse of the Rana Plaza building epitomizes the poor working conditions of the RMG sector in Bangladesh, which broadly includes, insufficient installation of safety measures, violations of safety and security measures, lack of emergency exits and adequate staircases, inadequate fire control management, lack of comprehensive emergency industrial accident management systems, improper safety inspections, avoidance of the Factory Building Act, and other compliance-related issues, including a failure to enforce national building codes [1,8,12,13,16,17]. As a result, the collapse of factory buildings has become a prime concern for the RMG workers because many of the RMG factory buildings in Bangladesh are not compliant with building and construction codes, and do not meet minimum standards including fire safety measures [8,13,18,19]. 

In relation to Rana Plaza, the owner of the factory building was illegally given permission by the local municipality to construct additional floors and construction work was underway to add floors to the top of the building at the time of the collapse [13,20], resulting in weakening of the overall structure. Since the collapse and associated investigations as to the cause of the collapse, it has been reported that the government was found to have failed in applying building codes and regular inspections [9,18,19,20,21] to the site. In addition, underlying corruption was found to have influenced government approvals due to the strong affiliations between the RMG owners and political powers in Bangladesh [9,17]. Even though these factory owners are characterized as the ‘power wielder’, able to continuously violate laws, rules, regulations, and agreements, government officials remain reluctant to take action against them [22].

In addition to sudden disasters such as the collapse of one factory building and fire in another factory, the overall working conditions in the Bangladesh RMG sector remain the worst among South and Southeast Asian countries where such manufacturing factories are prevalent [23]. These poor working conditions create environments where employees’ physical and psychological health are vulnerable. These workers face distinctive types of health vulnerabilities mainly due to the hazardous and unhygienic working conditions, long working hours without adequate breaks (e.g., RMG worker usually 9–12 h per day and 12–16 h work during peak times), irregular pay and low wages, and the nature of the work which is largely manual and includes repetitive tasks [23,24,25,26].

Notwithstanding these dangers and vulnerabilities, the RMG sector of Bangladesh creates a unique opportunity, providing around 4 million workers with employment, most of whom are women (approximately 80%) from rural backgrounds, with no or little education. Further, this sector is considered as the highest contributor to the GDP of Bangladesh and ranks in the top position in terms of export earnings (being that the RMG sector contributes 81% of the total export earnings) of the country [25,27]. This important sector is therefore situated in a power position in the political and economic environment of Bangladesh, enabling corruption and abuses of power which contribute to the dangerous conditions in which these workers are employed. 

Adding to these issues, the post disaster management in Bangladesh is inadequate due to lack of proper compensation, inadequate or inaccessible healthcare facilities, and the slow rehabilitation process to accommodate the survivors of disasters within the mainstream society [13,28]. For example, in the six years since the Rana Plaza collapse, anecdotal evidence suggests that survivors’ recovery from physical and psychological vulnerabilities have been negligible. This is largely due to inadequate financial compensation [9], lack of proper healthcare facilities and rehabilitation [29], and limited job opportunities in the RMG sector as physical strength is a primary requirement for this labor intensive job [28]. The consequences of the Rana Plaza collapse have resulted in significant adverse health outcomes and financial crisis for these already highly vulnerable workers.

Given the scope of the event and limited research on the topic, this study was conceived to better understand the impact on the health and wellbeing of the survivors. Findings from the study will provide insights to raise awareness of the workplace issues in the RMG industry, to help inform future prevention activities, and to ensure survivors needs are exposed to draw attention to adequate compensation and provision of appropriate healthcare services. As a result, the objectives of this study sought to identify the ongoing physical and psychological health vulnerabilities of the Rana Plaza survivors from their perspectives. It also examines access to, and uptake of, health treatment and compensation from both government and non-government organizations.

## 2. Materials and Methods 

A qualitative research design was applied in this study. Data were collected through in-depth, unstructured interviews with 17 Rana Plaza survivors (female: 11, male: 6) from April to July 2018 through the use of a pre-developed interview guide. Interviews were tape-recorded and transcribed verbatim. This study was conducted and reported in accordance with the consolidated criteria for reporting qualitative studies (COREQ) [30]. 

### 2.1. Participant Selection

Survivors of the Rana Plaza collapse, aged 18 or above, were selected as the participants for this study. Purposeful and snowball sampling techniques were applied to recruit participants in the study. The participants were reached via three contact persons (one contact person was an RMG trade union leader, the other two were the leaders of Rana Plaza Survivors’ Association). Since the first author had previous experience conducting research with Rana Plaza survivors [28,31], regular communication with on-site contact persons, who made phone calls to potential participants to provide information about the study, was utilized. Twenty individuals were approached to participate in the study, of whom 17 consented to participate (a response rate of 85%). The resultant interviews led to data saturation, so no further participants were recruited. 

### 2.2. Interview Procedure

The interview schedule was designed by H.K., M.M., M.S.I., and K.U. with the aim to explore (i) current socio-demographic characteristics of the survivors; (ii) factors that influenced/forced them to join in work on the day of the collapse; (iii) the immediate and the ongoing impact of the collapse on their physical and psychological health; (iv) the nature, duration, and impact of healthcare treatment on their physical and psychological health; (v) the current status of their physical and psychological health; (vi) the immediate and ongoing forms of financial or other compensation and from whom; and, (vii) the current status of financial or other compensation and from whom. The interview schedule was pilot tested with two Rana Plaza survivors. No adjustments were necessary following the piloting of the interview schedule. All interviews were conducted by the first author at a pre-arranged time in a mutually agreed location in the Savar Upazila administrative area. This location was chosen as it is near labor union offices (which is an appropriate location for female participants to attend) with tea stalls with open spaces nearby (which is an appropriate location for the male participants). Prior to the interview commencing, the entrances to these locations were closed to maintain privacy for the duration of the interview. 

Before commencing the interview questions, the participants were given a detailed description of the purpose and objectives of the study, including the position of the interviewer. The participants were also informed that the interviews would be conducted confidentially. The first author read the consent form to the participants and upon agreement, each of them signed the informed consent form. The interview commenced with general questions about the participants working and personal life and how they came to be employed in the RMG sector to assist with building rapport. Then open-ended questions with additional probing questions were used to further explore the topics of interest. The main topics covered in the interview were factors associated with the collapse, impacts of the collapse, and current overall status of the survivors. All participants were interviewed once and in the presence of at least three persons, the interviewer, and an intermediary person (preferably an RMG trade union leader/leader of Rana Plaza Survivors’ Association), who managed the participant for the interview, and a family member/friend of the researcher to avoid any perceived vulnerability produced during the interviews. Permission was taken from each of the participants about the presence of the external members in keeping with cultural expectations in Bangladesh. 

All the interviews were conducted by the first author, who is a current male PhD student at the University of New England (Australia), examining the health vulnerabilities of the RMG workers of Bangladesh. All interviews were audio recorded, and field notes were taken. The duration of the interviews ranged from 40 to 65 min. None of the researchers had any previous contact with the participants nor represent an organization or healthcare profession involved in health vulnerabilities of the Rana Plaza survivors.

### 2.3. Data Analysis 

Audio recordings were transcribed into Bangla and then translated into English by the first author as Bangla is the first language of the interviewees and the interviewer. Both English and Bangla scripts were then checked against the field notes as well as audio recordings and attested by a PhD fellow at the University of New England (Australia) who has expertise in both Bangla and English language, to ensure the accuracy of the data. At the end of each interview, the interviewer checked for shared meaning of the main issues of the interview. This was undertaken as it was not possible to share the written transcripts with the participants due to low literacy of the participants and logistical issues (interviewer returning to Australia to transcribe interviews, and participants having no postal address nor the ability to post back any amendments). 

Following transcription, data were reviewed by the researchers H.K. and M.M., and a thematic analysis was performed to identify the themes. Specific themes (which consist of words, sentences, phrases, paragraphs or even entire documents) are used to interpret the data obtained through the in-depth interviews. The theme identification process, which used an interpretative lens, occurred in three rounds: the first review focused on reading and becoming familiar with the content, the second review was to identify themes, and the third review was to apply the theme categories once they were refined [32,33]. Following on from the development of the thematic schema, Haddon’s Matrix (a paradigm/theoretical framework best known in the injury prevention field to arrange events into a temporal sequence) [34] was utilized to further conceptualize the temporal range of the events in the participants’ narratives; that is, the themes were then organized into a pre-event, event, and post-event framework.

### 2.4. Ethics

Approval was obtained to conduct this study from the Human Research Ethics Committee, University of New England, Australia (approval number: HE17-277), on 20 December 2017. All participants were given a pseudonym to protect their identity. 

## 3. Results

The participants were predominantly from rural backgrounds (i.e., only participant P originated from the urban area), aged between 19 to 43 years at the time of the interview. Fifteen out of 17 participants’ literacy level was up to primary school (class 1–5). In addition, participant D was a high school (class 6–10) student and participant P completed a Master’s degree from a college. All the participants were suffering from multiple physical (i.e., bone injuries/fractures, severe headache, eyesight problems, kidney problems, movement difficulties, and bodily pain) and psychological (i.e., trauma, fear, sudden anger, suicidal ideation, stigma, depression, sleep disorders, the feeling of insecurity, and nightmares) health issues. Regarding the availability of free health care facilities for follow-up treatment, only one participant (F) reported that she had received limited free healthcare facilities for follow-up treatment at the time of interview (Table 1).

In terms of current employment in the RMG sector, five participants mentioned that they had limited access to this sector, however, none of them were engaged in RMG work. The fact they were not working in the sector was reported as related to their fear of working in this sector or because they had decided to change their occupation after the disaster. On the other hand, 12 participants described that they no longer have access to employment opportunities in the RMG sector. Some of them joined RMG work following the Rana Plaza collapse but later they quit due to the adverse conditions they experienced including becoming unable to fulfill the daily production target fixed by the management, intolerance of the sound produced from the sewing machine, or inability to employ the full physical strength which resulted in verbal abuse by the management. Ten of the 17 participants found new opportunities through running small businesses such as grocery shops/tea stalls, pharmacies, and working as a mason and salesman. However, seven of the participants (females: 5, males: 2) reported being unable to find a suitable job because of their worsening physical conditions. Regarding compensation, 14 participants had serious complaints about receiving insufficient compensation and expressed their disappointment with the amount received from the government and international brands. Moreover, they emphasized the economic hardships and urged for urgent steps be taken by the government, Bangladesh Garment Manufacturers and Exporters Association (BGMEA), and the international brands for proper compensations and rehabilitation. In addition, all the participants believed that as their issues were becoming more distant, they expressed fear that their tragic history would be soon forgotten (Table 1).

Based on the interviews, the following themes (Table 2) were constructed using Haddon’s matrix [34]. The themes and sub-themes are discussed broadly under this matrix which comprises of three items: (i) pre-event—which includes the situation before the collapse of Rana Plaza; (ii) event—the event at the time of Rana Plaza collapse and the immediate rescue; and (iii) post-event—focuses on the issues since the collapse of Rana Plaza.

### 3.1. Pre-Event (Before Rana Plaza Collapse)

#### 3.1.1. Aspects Related to the Collapse of Rana Plaza

##### Ignoring or Providing False Information about the Crack Found in the Rana Plaza and Safety of the Factory Building

The collapse of Rana Plaza occurred on 24 April 2013. The day prior to the collapse, there was a significant pre-event, where a crack was found in the building [1]. Due to the significant nature of the crack, some tenants of the Rana Plaza chose not to open their businesses on the day of the collapse. However, the RMG factories continued to operate. The management downplayed the reality of the crack and the safety of the building so that the workers would continue to work on the day of the collapse. The pressure of international buyers to receive their goods on time (for example, the shipment of warm clothing before the start of winter), put pressure on the sourcing agents or buyers (that is those in the middle position between the international brands and the clothing factory owners), who ultimately put pressure on factory owners and their employees. These pressures did not ease when the building was found to be structurally unsound following the identification of a crack, and factory employees were provided misinformation, as this participant explains:


*“We were given release from work at 11 a.m. during the day before the collapse. Our general manager (GM) sir announced that there was a simple electrical problem and that the electrician was working to fix it. He advised us to come back to work on the following day” *
(L, Sewing operator)

Participants were aware that the situation was unusual, as they had been given unscheduled time off the day prior to the collapse, yet they were told to continue working on the day of the collapse, as this participant graphically explains:


*“Early morning of the day of the collapse, I contacted the production manager (PM) sir of my factory via mobile phone to know whether I should go to join in work. He told me to go there. He further added, ‘if we remain alive, we will survive unitedly; and if we die, we will die unitedly.’”*
(I, Sewing operator)

The failure of the management to close the factory, until it was safe, put the workers at great risk of harm. It is apparent these actions were for pecuniary interests; that is, they tried to justify that the factory building was safe for the workers as they did not want to stop production. This quote demonstrates that the PM was aware that there were inherent risks in attending work on the day of the collapse. Nevertheless, workers were encouraged to return to work, as the following participant recalls information about the state of the building:


*“The management of my factory confirmed us saying that ‘the engineer has checked the building. There is nothing to be worried.’”*
(N, Line chief)

Furthermore, where workers voiced their concerns, they were not taken seriously:


*“The PM of our factory told us ‘go to the 3rd floor and watch whether the workers are working on that floor, where the crack was seen at first. If they can work, why cannot you?’…The authority ensured that the building was safe to continue working.”*
(K, Sewing operator)

The actions undertaken by management of these factories on the day prior to, and day of the collapse, along with policies that financially penalize those who are unable to attend the workplace, resulted in many people being inside the buildings when the collapse occurred. Yet, the recollection of the worker saying that their manager mentioned that if they are to die, they will all die together, indicates that those in positions of authority were aware of the dangers of attending the building on that day.

To sum up, the ‘pre-event’ phase articulates the false information regarding the safety issues of the Rana Plaza building that was spread among the workers by the factory management, which resulted in additional risks for individuals working during the day of the collapse. 

### 3.2. Event (During Rana Plaza Collapse and Rescue)

#### 3.2.1. Experience of the Day of Collapse and the Immediate Rescue Response

In Haddon’s Matrix [34], ‘Event’ incorporates the experience of the workers on the day of the collapse, why they entered the factory building, the experience of being trapped inside the factory, and how they were rescued. The compulsion to attend the workplace for the participants was two-fold. First, the need to remain employed related to their own needs and those of dependent family members in rural areas, motivated the participants to attend work even though they feared the conditions were not safe. Second, these factories pay an attendance bonus to employees who attend work each day of the month. As the damage to the building was discovered toward the end of the month, employees were concerned that if they did not attend work on that day, they would be financially penalized. These concerns are echoed below:

##### Force and the Components which made the RMG Workers Join in Work

Along with the pressures to attend work, there were other reasons the workers attended on the day of the collapse such as fear of losing their job, month-end salary, and work presence bonus. 


*“The supervisor of my factory forced me to join in work. I knew if I did not go to work on that day, I will not be paid the monthly salary. I was supposed to receive the salary of the running month just after 10 days. Ensuring the salary at the end of the month was more important than thinking about the probable outcome of joining in work on the day of the collapse.”*
(C, Sewing operator)


*“Not only had the factory management, but the owner of the factory building also threatened us to join in the work. Moreover, the management played music system loudly so that we become unable talking to each other about the crack of the building and to continue working without any fear.”*
(G, Sewing operator)

Some of the workers joined in work willingly so that they could claim the work presence bonus with the month-end salary. The poor socioeconomic condition of the workers thus influenced their decision to work on that day. Participant H clarified that they receive a presence bonus, also known as *hajira* (presence) bonus, and even though this bonus is generally a very small amount (varies from approximately USD $12–15 for one month), it is an important addition for these low income earning workers; consequently, they try not to take any sick leave or days off to achieve this bonus, as explained below:


*“Usually, if we do not remain absent from work in a month, we are rewarded with the presence bonus. I did not want to lose the opportunity to get the presence bonus along with the running month salary and joined in work willingly on the day of the collapse knowing about the crack.”*
(H, Sewing operator)

##### Evacuation System inside the Rana Plaza Building

When the collapse occurred, many people were inside the building and those not killed in the collapse needed to escape the building. However, as Participants A and H explained, the evacuation systems that existed inside the building were inappropriate or inaccessible to the employees when they needed to escape: 


*“The front door of the factory was always kept locked so that the workers can’t go outside. And the emergency exit gate was also blocked by the clothing stuff. I think if there were enough emergency exists available in the building and the front door was kept unlocked, many of my fellow workers could survive.”*
(A, Quality inspector)


*“The building had only one gate and the stairs were very narrow…The workers could not use the gate to escape.”*
(H, Sewing operator)

Thus, the ‘event’ epitomizes issues regarding the day of the collapse, including the influence of the factory management or building owner to encourage attendance at the workplace on the day of the collapse including the threat of no month-end salary with presence bonus and a poor evacuation system that existed inside the building.

These conditions ultimately resulted in many people (around 5000) remaining in the building at the time of the collapse, due to inadequate escape opportunities when the first sign of collapse occurred. Those who were able to escape or survived within the building have been left traumatized and with significant physical and psychological scarring, as this participant describes: 


*“Today I think it was better if I was found dead during the collapse, surviving with different health complexities along with the jobless, life has become meaningless for me.”*
(M, Iron man)

### 3.3. Post-Event (After Rana Plaza Collapse)

The ‘post-event’ phase details the health vulnerabilities of the participants, the structural failure of the relevant agencies to appropriately respond, and the self-motivation found by some of the survivors to live their lives, albeit in new ways. While the individual participant is affected by the collapse, especially in terms of physical and psychological health vulnerabilities, the structural failure of the state and BGMEA can be considered as a factor in the current vulnerable socioeconomic and political conditions of the survivors.

#### 3.3.1. Physical Health Vulnerabilities/Status of the Rana Plaza Survivors

The interviews for this study were undertaken six years after the collapse of Rana Plaza. Thus, participants have experienced a significant period to incorporate the event into their lives, and adjust to the limitations placed upon them by this trauma. Yet, these interviews demonstrate that many individuals remain vulnerable to different types of physical and psychological issues as a result of the event. The study identified some common health issues (such as bone injuries/fractures and amputation, head injury and eyesight problems, and kidney problems), which created long-term health vulnerabilities. These are explored further below. 

##### Bone Injuries/Fractures and Amputation 

The most common physical health complaint among the participants was related to bone injuries/fractures which includes mainly backbone, legs, and hands. Where damage to limbs had been experienced, these were considerable, and required significant medical intervention, as this participant describes: 


*“One of my legs was broken due to the fall of my sewing machine...Doctors had poured an artificial rod into my waist. My rib cage was also broken.”*
(F, Sewing operator)

These types of injuries were common and have resulted in ongoing issues, as this participant describes: 


*“My backbone is now a half inch shorter and three bones are decayed. In addition, my left hand was broken due to the fall of part of the roof.”*
(B, Sewing operator)

In some cases, the fractures became so serious that some of the survivors had to go through major surgery to amputate limbs, including hands and legs. Such amputations end the career of the RMG worker, given this work is manual labor, as this participant details: 


*“My right leg got fractured by the collapse. … I was admitted to the Enam medical college and hospital where doctors decided to amputate my right leg. I can’t move naturally like before.”*
(L, Sewing operator)


*“I lost my right hand by the collapse. Now, I depend on my left hand to do anything…I cannot explain how I feel about losing my hand.”*
(P, Sales representative)

In addition, these workers are not economically well-off and are not able to hire someone to look after them in the given situations. Thus, health vulnerabilities also led to socioeconomic vulnerabilities. Participant H clarified the issue:


*“My husband died in 2009 in a road accident. I took the responsibility of my family and joined in RMG work. As I am not capable to do work now, we are living in extreme poverty… I wish to have economic help hand to hand. And I demand to receive the pension for the rest of my life.”*
(H, Sewing operator)

##### Head Injury and Eyesight Problems

While limb injuries were common, so too were injuries to the head and eyesight. These originated from the collapse of the roof and/or heavy materials and the resultant dust and debris in the environment. Participants reported a range of symptoms, including short-term memory loss and commonly severe headaches. 


*“I am suffering from an extreme headache. Within 20 minutes headache starts while talking as my head was injured by heavy things on the day of Rana Plaza collapse. I also cannot see anything from a distance.”*
(E, Sewing operator)


*“My head, face, and eyes were hurt by the fallen brick at the time of Rana Plaza collapse. I lost four teeth due to that injury. Every day I have to take a pain killer tablet for the headache.”*
(Q, Sales representative)

##### Kidney Problems

Another common health issue for the Rana Plaza survivors is related to their kidney function. Participants M and O explained how they came to know about their kidney problems and why they could not afford to continue treatment of the problem. 


*“Currently I am suffering from kidney problems. According to the doctor, as I was injured by heavy things during the collapse, my kidney got damaged.”*
(M, Iron man)


*“After the collapse, the doctors dialyzed my kidneys and suggested to go for kidney transplantations, I could not afford money for further step. Sometimes, I feel severe pain in my stomach now-a-days.”*
(O, Sewing operator)

In addition to the physical health vulnerabilities, the survivors were also suffering from psychological health vulnerabilities.

#### 3.3.2. Psychological Health Vulnerabilities/Status of the Rana Plaza Survivors

While some participants appear physically healthy and are outwardly leading a typical life, when describing their inner worlds, the damage from the Rana Plaza collapse became apparent. The participants described some key psychological issues such as nightmares, trauma, depression and suicidal ideation, and sudden episodes of anger.

##### Nightmare/Fear

Participants K, P, and F mentioned that they very often have vivid dreams about the collapse. 


*“Sometimes I dream my fellow workers who died in the collapse. Their faces are very clear to me till today and they want to say something. And suddenly I woke up with strange feelings in the midnight. I also get frightened if anyone touches me while I sleep.”*
(K, Sewing operator)

They reported being scared of repetitions of the same collapse again in their life. Since the participants witnessed many deaths of their fellow workers, it was hard for them to forget the memories of those people. Importantly, they are now fearful of working or living in buildings.


*“I dream bad dreams very often and become scared still today if I remember the day of the collapse. I cannot think of working or living in the high rise building as I am frightened of the repetition of the same incident again in my life.”*
(P, Sales representative)

These nightmares still do not give them the chance to live their lives without fear or forgetting the Rana Plaza tragedy. 


*“In my dream, some of them (fellow workers) want water from me and invite to go to their world (she meant about after death)… It was an unbearable hot condition inside the collapsed building and many of my colleagues were crying for water…and fallen to death.”*
(F, Sewing operator)

In this way, the participants are also struggling with ongoing depression and trauma which may ultimately force them to think about suicide.

##### Trauma, Depression, and Suicidal Ideation 

The collapse of Rana Plaza also destroyed the childbearing capacity for some women. This participant was pregnant at the time of the collapse, and lost her unborn baby:


*“I was pregnant and my child had died in my womb due to the fall of a beam on my belly/abdomen. According to the doctor, I will not be able to conceive a baby anymore. I cry often thinking about my unborn baby.”*
(E, Sewing operator)

For others, the injuries caused by heavy items such as beams, fallen roof trusses, machines, bricks, rods, falling on them resulted in them being unable to carry a pregnancy in the future:


*“The doctor confirmed me that I will not be able to give birth because my belly/abdomen got seriously injured by the collapse. This incident is the saddest part of my life. I do not find any meaning of my life….”*
(C, Sewing operator)

The lack of being able to make meaning from these injuries has resulted in depression, suicidal ideation, and reports of survivors ending their lives. Participants indicated that due to the conditions (after the Rana Plaza collapse) including the inability to have children, health complications and injuries, anxiety about the future, isolation, and fear of becoming a burden on the family, made them vulnerable to suicidal ideation. Reports, such as anecdotal ones made by the president (A) of Rana Plaza Survivors’ Association (personal communication, 2018), suggest that individuals who survived the collapse have gone on to die by suicide (reportedly in 2015 and 2016). Among the participants in this study, there was clear evidence of suicidal thinking, such as this participant details: 


*“I think it could be better if I found dead on the collapse…It is painful to be treated as a burden by own family.”*
(B, Sewing operator)

With regard to suicide attempts:


*“It was only me, who survived from the collapse among 13 workers working in a single room. I cannot walk without a crutch now. My wife left me after the collapse which made me upset and I attempted to suicide at that time.”*
(A, Quality inspector)

##### Sudden Anger

Another important psychological issue prominent among the participants was a lack of emotional regulation as a result of the collapse, where participants explain they become angry without a discernable reason.


*“I have become short-tempered now. I talk too much than before. If I get angry with anyone, be that for a simple reason, I want to hit him. If I can’t hit him, I hit myself.”*
(J, Quality inspector)

This may be exacerbated by the avoidance of other people and general stigmatization of being a Rana Plaza survivor that may result in this loss of emotional control, as this participant explains:


*“People drive me away when they get to know that I am a Rana Plaza victim… I cannot tolerate this… and become angry instantly.”*
(O, Sewing operator)

#### 3.3.3. Structural Failure to Manage the Post Rana Plaza Tragedy

Along with the other disadvantages, the participants shared the barriers to accessing free healthcare facilities for follow-up treatment (which made them more vulnerable to further health issues) they experience along with issues regarding re-employment opportunity to the RMG sector.

##### Current Access to Healthcare Facilities: Follow-Up Treatment Facilities

Participant F mentioned that although she had access to free healthcare facilities, she could not avail herself of those facilities due to the lack of transport and difficulties with movement due to her injuries:


*“I feel severe pain in my neck. Doctors suggested me to take regular therapy, which was initially provided by a government hospital in free of cost. But who will give me the conveyance (bus/rickshaw fare) and company to go and come from the therapy center?”*
(F, Sewing operator)

In addition, participant Q mentioned that he stopped going to the hospital for follow-up treatment due to the pressures of daily work and unavailability of free healthcare facilities.


*“Due to the expenses of treatment and huge work pressure, I can’t go to the hospital nowadays for a regular follow-up treatment and physiotherapy. And I do not get free treatment from anywhere now although I used to receive till 2015 from the GONOSHASTHAYA KENDRA (Peoples Health Centre).”*
(Q, Sales representative)

While some did have access to free health care, this was not experienced by all. Others reported they were subsequently ignored by the management of the hospitals from where they received free treatment immediately after being rescued from the collapse. For example, participant C shared her experience of being rejected by the hospital management when she went there for follow-up treatment.


*“I was under treatment in the CMH (combined military hospital, Savar) for 13 days after being rescued from the collapse. After those 13 days of treatment, I did not receive any treatment from anywhere. Once I went to the CMH for treatment purpose but the CMH authority even did not let me go in for further follow-up treatment.”*
(C, Sewing operator)

##### Lack of Support from the BGMEA/Factory Owners: Survivors’ Access to RMG Sector

Following the collapse and any rehabilitation, some participants reported wanting to find new employment within the RMG sector. However, as participants I, J, and E all reported, they had ‘limited’ or ‘no’ access to this sector because the factory owners treat them as less productive due to the injuries they sustained in the collapse.

Participant I reported that being a Rana Plaza victim did not result in any accommodations being made in her employment now, to allow for her injuries, from the factory management when she started to work again.


*“After the collapse, I faced problem to get the job in the RMG sector as they (manager/the employer) thought I will not able to employ full physical strength to fulfill daily work target. Later I got the opportunity to work as a sewing operator in a garment factory. Since I could not finish my work on time, the PM used to scold me. I left that job.”*
(I, Sewing operator)

Participant J expressed interest to work in the RMG sector, unfortunately, he did not get an opportunity to work again in this sector: 


*“No factory owner wants to give me a job when they get to know that I am a Rana Plaza survivor. I want to join in this sector again because I am not skilled at other works.”*
(J, Quality inspector)

While participant E appealed for an easier task and flexible hours:


*“I think I will be able to work in this sector again if the factory management gives me the opportunity to do easy tasks and flexible hours. It would be tough for me to work for long hours (i.e., 10 to 12 h, including the overtime work) like before.”*
(E, Sewing operator)

##### Compensations and Rehabilitation Progress

Overall, all the participants reported being unsatisfied with the role of the government regarding compensations and rehabilitation progress. As demonstrated by this participant who discussed the small compensation they received following the collapse.


*“I received BDT 50,000 ($625) from the Prime Minister’s Trust Fund (previously known as Rana Plaza Trust Fund), which was mostly spent on buying medicines and food on those days. Later, I received BDT 45,000 ($560) through three transactions. I do not know the exact source of this amount…probably from Walmart. I did not hear anything yet about more compensations or whether the government will take any proper step to rehabilitate us… Now my economic condition is the worst. Sometimes, we have to struggle for food to eat three times a day.”*
(B, Sewing operator)

However, others did not receive compensation, even though they believe this is a responsibility of the government towards its citizens:


*“I did not receive any kind of financial help from the government. Nobody from the government yet contacted me. Being a citizen of Bangladesh, I want the government to take my responsibility.”*
(D, Helper)

While some did report receiving compensation, this was commonly used for minimal health care needs and did not take into account the long term care needs, nor the lack of future employment security. These sentiments appear in contrast to how participants understood compensation and health care needs would be met. The participants expressed doubt whether some non-government organizations (NGOs) (through which international brands and other international humanitarian organizations usually prefer to disburse financial aid) received aid but did not disburse it among them. Participants questioned how these monies were handled and what happened to compensation money that was understood to be directed toward the survivors, as these participants explain: 


*“…many NGOs gave us the hope to settle down permanently but they did not do anything rather we doubt that many of them made money from international organizations by selling our miserable stories.”*
(G, Sewing operator)


*“Many people from different organizations gave me hope for the financial help and took my identifications which included names, photos, signatures, and medical certificates. But nobody came to me again.”*
(O, Sewing operator)

#### 3.3.4. New Hope

##### Motivation to Live 

While many stories the participants told were of horror, trauma, and injury, there were also stories of hope, where some survivors described being motivated to find new meaning and purpose in their lives. For some, this was through starting a new small business with the small compensation payments they received, or by selling their lands, and taking loans from close relatives as the following participant explains:


*“Currently, I am running a small grocery shop. I do not want to live my life depending on the KORUNA (grace) of others. Please pray for me so that I can run the shop very well.”*
(N, Line chief)


*“I started a small pharmacy with the small amount of money received from the Rana Plaza Trust Fund. As I am not capable of doing jobs where physical labor necessitates, I am surviving the little income comes from this pharmacy. I have a plan to extend this pharmacy in the future.”*
(A, Quality inspector)

In addition to the independent businesses started by some of the participants, participant D shared that she started school with a dream of pursuing higher studies and a better, descent life with her family.


*“Now I have started to go to the school again and enrolled in class eight. I want to finish my school and college with a good result so that I can get admission to a good university of Bangladesh. After finishing graduation, I want to contribute to my family by doing a job.”*
(D, Helper)

Finally, the ‘post-event’ theme describes the ongoing physical and psychological health vulnerabilities, along with the overall wellbeing of the participants. It also focuses on the participants’ limited access to healthcare facilities and re-employment in the RMG sector, underlying which is the failure of the relevant agencies to ensure safety and security for those injured.

## 4. Discussion

The study identified some key factors (i.e., force, threat, and fear to lose the month-end salary and presence bonus) to explain why the Rana Plaza workers decided to attend work on the day of the collapse, even when they were aware of the dangers present in the building. We found that even though the workers were aware of the dangers, they attended work on the day of the fateful collapse because of issues such as the powerful factory bosses, fear of the loss of their job, concern they would lose their monthly attendance bonus, and due to the family in rural areas who depended on them for financial assistance. A previous study [28] also found that Rana Plaza workers were forced and threatened to work on the day of the collapse. Another study identified RMG workers as less capable of bargaining and in a more vulnerable position because of being uneducated, unskilled, unaware of labor rights, and in the fear of losing little monetary contribution to their families [35], which may have led to Rana Plaza workers being at work at the day of the collapse.

The study contributes to current understandings of the ongoing health vulnerabilities of those who survived the building collapse. We found that the participants were suffering from different types of physical and psychological health issues. In addition to the physical health issues, such as bodily pain, movement difficulties, headache and eyesight problems, and kidney problems, the participants were found to be living with many types of psychological health issues including reduced future potential life events, including child-bearing, and posttraumatic stress due to recurring memories of the collapse. Similar results have been reported in other studies of Rana Plaza survivors conducted closer to the time of the disaster [36,37], and reflect the physical and psychological health vulnerabilities reported by other natural disaster events [38,39,40,41,42]. As a result of the psychological trauma experienced by the survivors, many reported ongoing thoughts of suicide and remain a potential risk of suicide in the future. Previous research supports the finding that traumatic events and symptoms of vulnerable psychological health such as depression, extreme grief, anxiety, and obstacles to leading a normal life, have the potential to lead to suicidal deaths [28,43,44,45]. This study further revealed that the Rana Plaza building lacked enough available safety exits and stairs, and the exits that were available were blocked by clothing materials limiting the number of people who could escape the building leading to an increase in loss of life. The survivors reported that knowing this added to their ongoing trauma from the event, and some clearly described what is known as ‘survivor guilt’ [46], and is particularly evident in those who report they wish they had died on that day. Participant stories demonstrate the failure of the relevant agencies of the government to monitoring the safety regulations relevant to the Rana Plaza building and to ensure the extension was adhering to the ascribed building codes. Since Rana Plaza was not built in accordance with the proper building codes and lacked emergency exit plans [8,9,28], this collapse resulted in more loss of life than any other such tragedy in the RMG sector.

Participants in this study also reported barriers to receiving free health care facilities for follow-up treatment for the injuries they received, and lack of any (in some cases) or sufficient compensation, due to the injuries sustained as a result of the Rana Plaza building collapse. The study also revealed that survivors were either reluctant to join in RMG work or there was no accessibility to this sector for them anymore. It is reasonable for workers to expect adequate health care and compensation for injuries received in an event such as the Rana Plaza collapse [28]. However, for many of the survivors interviewed in this study, compensation and adequate health care were far from reality. The majority of the participants mentioned that they received free healthcare facilities immediately after the Rana Plaza collapse, however, at the time of conducting interviews, they reported being refused in terms of receiving follow-up health care facilities by the previous healthcare providers. The study also identified that the compensation received by the participants was negligible compared to the post Rana Plaza economic hardship that they faced. A study conducted by Fitch et al. [37], showed that 83.4% of the Rana Plaza survivors remained unemployed, and 57.3% reported receiving a quarter or less what they were promised as compensation. The government, along with the international brands (who used to buy clothes from the factories located at Rana Plaza) and BGMEA, should take urgent steps to ensure lifelong healthcare facilities and adequate compensation is provided for the survivors. The findings from the study suggest the need for accessible, lifelong physical and mental health care for Rana Plaza survivors. In light of the physical and mental health complaints experienced by the participants in this study, the impact on their ability to undertake paid employment, and the health care systems in Bangladesh, any services would need to be free of charge and local to the residence of the survivor. Further, the compensation that has previously been promised needs to be provided in order for them to survive in mainstream society in a solvent way financially.

Signing the ‘Accord on Fire and Building Safety in Bangladesh’ on 15 May 2013, by the international clothing brands and retailers as a consequence of the Rana Plaza collapse, is probably the biggest achievement to result from the collapse to ensure the ongoing safety of the workers in the Bangladeshi RMG sector [5,6]. Through this Accord, 80,000 hazards were identified in the RMG sector and 32 RMG factories closed after being found to be unsafe [7]. However, the continuation of this Accord, while it has the ability to increase safety, is in some doubt. The original timeframe for the Accord has ended (originally for five years ‘May 2013 to May 2018’ and later they were given an extension for six months as the interim period) [47] and without this international pressure, along with internal political issues, it is possible that factory workplace conditions will decline. Should this occur, another disaster, such as the one experienced at Rana Plaza, is inevitable. Thus, continued pressure to regulate this industry is required to ensure the safe work practices continue, and that those injured through such work are not discriminated against. This should be a shared concern among the community, the state, BMGEA, and all international brands commissioning work RMG work in Bangladesh. The findings of the study can be used to avoid the same kind of incident in a similar setup such as the South and Southeast Asian country perspectives, which house the majority RMG factories of the globe [23], and further protecting the daily activities of workers in this industry. 

### Limitations and Strengths 

Following the collapse, many survivors of Rana Plaza relocated to other regions due to the end of their employment, becoming unable to maintain the cost of living in an urban area and absence of receiving rehabilitation or compensation. This limited the number of potential participants available to participate in the study and impacted the recruitment strategy applied. In addition, because of the qualitative nature of the study, the findings need to be interpreted with caution. The study respondents were self-selected; therefore, results may be biased by age, sex, ongoing health complaints, employment status, access to free healthcare facilities for follow-up treatment, and desire for compensation. The health vulnerability issues were reported in accordance with the interviews; we did not follow any medical procedure (it lacked using any diagnostic instruments) to identify their existing health complaints. In addition, the study results were based on the memories of the survivors which may produce misleading information in some cases. Moreover, the results were solely based on the survivors’ perspectives/point of views; we did not collect responses from the others such as factory owners, sourcing agents of the international brands, the BGMEA authority, or members of the government.

However, these limitations can be considered within the strengths of the study. First of all, the strengths of this study include the range of ongoing physical and psychological health vulnerabilities identified, available healthcare facilities and employment opportunities, and the role of the state, BGMEA, and international brands regarding compensation and rehabilitation. Second, the attestation of the transcriptions and translations (which were labeled with the audio recordings) was undertaken by a person who was expert in both Bangla and English language, to ensure the transparency of the themes/findings used in this paper. Third, the use of the Haddon’s matrix to analyze the themes, allowed for presenting the results of the study in a more meaningful way. Lastly, to our knowledge, this is the first qualitative study which focused on ongoing physical and psychological health vulnerabilities of the Rana Plaza survivors along with their access to RMG sector and free healthcare facilities and can be considered as the starting point for future research from broader a perspective.

## 5. Conclusions

We used Haddon’s matrix [34] to shape the themes of the research. The ‘pre-event’, ‘event’ and ‘post-event’ issues provided a background for our main focusing point of physical and psychological health vulnerabilities of the Rana Plaza survivors. Participants described their ongoing issues, including their health complaints, limited access to the RMG sector, the availability of free healthcare treatment facilities, and rehabilitation progress. Some of them expressed concern their issues had become too old to receive any of the benefits promised by the relevant agencies. Obviously, this event has left a marked impact on the lives of the survivors of the disaster. One participant summed this up as follows: she said, “*the Rana Plaza tragedy will alive as long as I stay alive. I cannot forget this incidence due to the worst impact of the collapse on my life*” (H, Sewing operator). It is important that the collapse is not forgotten and that the lessons learned from the event are heeded by the government, the BMGEA and the manufacturing companies that commission work in Bangladesh. While prevention of similar events is beyond the scope of this research, drawing attention to the human suffering that has occurred as an outcome of this event, may assist in consideration of workers needs before, during and after catastrophic workplace events such as the collapse of Rana Plaza building. However, in conclusion, we argue that structural issues with the building caused the collapse and the high mortality was caused by false or misleading information being provided to workers that ultimately endangered life. To avoid such a situation in the future, strict inspections of the structural feasibility of factory buildings along with the regular monitoring of safety and security related issues by an independent inspection entity or authority (such as ‘Accord on Fire and Building Safety in Bangladesh’) will reduce the repetition of such type of catastrophic collapse in future. 

## Figures and Tables

**Table 1 ijerph-16-02342-t001:** Characteristics of the Rana Plaza survivors by existence health complaints, access to free healthcare facilities, current job status, and compensations.

Participant	Age	Sex	Literacy Level	Existing Physical Health Complaints	Existing Psychological Health Complaints	Current Access to Free Health-Care Facilities	Current Access to RMG Sector	Current Job Status	Compensations Received (Approx. in USD)
A	33	M	Primary school	- Bone injury - Broken ribcage - Headache - Movement difficulties	- Depression - Sleep disorder - Trauma - Suicidal ideation	No	Limited access	Running a small pharmacy	$1900
B	37	F	Primary school	- Backbone pain - Headache - Broken hand - Thyroid (throat) infection	- Nervous breakdown - Scared - Feeling of insecurity of living alone - Considering life as meaningless	No	No	Running a small grocery shop	$1180
C	24	F	Primary school	- Chest pain - Headache - Ligament injury of ankle - Movement difficulties	- Trauma - Suicidal ideation	No	Limited access	Unemployed	$275
D	19	F	Admitted to class VIII	- Bodily pain - Headache - Eyesight problem	- Sleep disorder	No	Limited access	Unemployed	$2300
E	30	F	Primary school	- Headache - Eyesight problem - Unable to talk for long time	- Trauma - Sleep disorder	No	Limited access	Unemployed	$1180
F	40	F	Primary school	- Broken legs and ribcage - Pain in neck - Headache - Movement difficulties	- Nightmare - Scared of repetition of collapse of building - Feeling of insecurity of living alone	Limited access	No	Running a small grocery shop	$1100
G	26	F	Primary school	- Head and leg injuries - Bodily pain - Headache	- Trauma - Frustration	No	Limited access	Started a small grocery shop	$1180
H	38	F	Primary school	- Backbone injury - Pain in leg - Headache - Movement difficulties	- Trauma - Depression	No	No	Running a small grocery shop	$2400
I	28	F	Primary school	- Pain in spinal cord, waist, and chest	- Nightmare - Scared of loud sound - Trauma	No	No	Started a small business	$1180
J	34	M	Primary school	- Broken hand	- Sudden anger - Depression	No	No	Work as a mason	$2200
K	31	F	Primary school	- Broken hand - Pain in head and chest - Breathing problem	- Nightmare - Considering life as meaningless	No	No	Unemployed	$1300
L	22	F	Primary school	- Lost one leg by the collapse - Movement difficulties - Backbone pain	- Sleep disorder - Anxiety - Depression	No	No	Unemployed	$1700 (She also received $12,500 as a Family Savings Bond)
M	25	M	Primary school	- Kidney problems - Headache - Bodily pain	- Scared of high rise building - Trauma - Frustration - Depression	No	No	Unemployed	$1180 (He also received $8100 as a Fixed Deposit)
N	39	M	Primary school	- Kidney problems - Broken hand and leg - Movement difficulties	- Sleep disorder - Scared of loud sound - Trauma	No	No	Running a small grocery shop	$3200
O	38	F	Primary school	- Pain in legs and waist - Headache - Movement difficulties	- Sudden anger - Scared of loud sound - Stigma	No	No	Running a tea stall	$4100
P	35	M	Master’s degree	- Lost one hand by the collapse - Movement difficulties	- Nightmare - Depression - Stigma	No	No	Unemployed	$1775 (He also received $15,000 as a Fixed Deposit)
Q	43	M	Primary school	- Headache - Eyesight problem - Kidney problems	- Scared of loud sound - Depression	No	No	Salesman	$1300

**Table 2 ijerph-16-02342-t002:** Summary of themes.

Phases	Themes	Sub-Themes	Factors
**Pre-event** (Before Rana Plaza collapse)	Aspects related to the collapse of Rana Plaza	Ignoring or providing false information about the crack found in the Rana Plaza and safety of the factory building	Factory management
**Event** (During Rana Plaza collapse and rescue)	Experience of the day of collapse and the immediate rescue response	Force and the components which made the RMG workers join in work Evacuation system inside the Rana Plaza building	Factory management/poor socioeconomic conditions of the workers, and physical environment of factory building
**Post-event** (After Rana Plaza collapse)	Physical health vulnerabilities/status of the Rana Plaza survivors	Bone injuries/fractures and amputation Head injury and eyesight problems Kidney problems	Human (individual)
	Psychological health vulnerabilities/status of the Rana Plaza survivors	Nightmare/fear Trauma, depression, and suicidal ideation Sudden anger	Human (individual)
	Structural failure to manage the post Rana Plaza tragedy	Current access to healthcare facilities: follow-up treatment facilities Lack of support from the BGMEA/factory owners: survivors’ access to RMG sector Compensations and rehabilitation progress	Socioeconomic and political environment
	New hope	Motivation to live	Human (individual)

The terms used in the table are modified versions of the terms originally used by Haddon [34].

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
