# Peer review of "The Current Health and Wellbeing of the Survivors of the Rana Plaza Building Collapse in Bangladesh: A Qualitative Study"

_ijerph, 2019, doi:10.3390/ijerph16132342_

Round 1
Reviewer 1 Report
This paper focused on investigating health and well-being among Rana Plaza Building collapse survivors using a qualitative method. I believe that the content of this paper can provide a lot of information on several issues, such as how to prevent the accident occurred again. However, some areas may need to be further concerned. My observation is shown as follows:
The authors should clearly state the objectives of this study. Also, they may need to further explain why this accident should be concerned and studied.
For the part of "Results", the framework of interviews had been shown. However, some key points should be summarised in each part. This can help the reader to easily understand the interview results.
I think the authors can further compare the conditions of survivors before and after the accident. This may provide more information for potential implications of the current findings.
For the part of "Discussion", it is no doubt that the survivors' physical and psychological health had been impaired after the accident. Therefore, the authors require to further explore whether new insights can be concluded from the findings. In addition, further implications for daily practices should be further elaborated.
I hope the above comment is useful, and many thanks for giving me this opportunity to review this manuscript.
Author Response
Answer to reviewer
The authors are very grateful to the reviewer and to the editor for their suggestions to improve the quality of the article.
This paper focused on investigating health and well-being among Rana Plaza Building collapse survivors using a qualitative method. I believe that the content of this paper can provide a lot of information on several issues, such as how to prevent the accident occurred again. However, some areas may need to be further concerned. My observation is shown as follows:
Response: Indeed, this research could include more information. However, recognizing the fact that every research has its inherent limitation, we have added an additional sentence at the end of ‘Conclusions’ section regarding your concern (how to prevent the accident occurred again) which now reads: ‘While prevention of similar events is beyond the scope of this research, drawing attention to the human suffering that has occurred as an outcome of this event, may assist in consideration of workers needs before, during and after catastrophic workplace events such as the collapse of Rana Plaza.’ (see ‘Conclusions’, Ln 655-658, P. 19)
Responses to the reviewer’s specific observations are given below:
1. The authors should clearly state the objectives of this study. Also, they may need to further explain why this accident should be concerned and studied.
Response: Following your suggestion, the objectives of the study and the explanation of why this accident should be concerned have been addressed clearly. This now reads: ‘Given the scope of the event and limited research on the topic, this study was conceived to better understand the impact on the health and wellbeing of the survivors. Findings from the study will provide insights to raise awareness of the workplace issues in the RMG industry, to help inform future prevention activities, and to ensure survivors needs are exposed to draw attention to adequate compensation and provision of appropriate healthcare services. As a result, the objectives of this study sought to identify the ongoing physical and psychological health vulnerabilities of the Rana Plaza survivors from their perspectives. It also examines access to, and uptake of, health treatment and compensation from both government and non-government organizations.’ (see ‘Introduction’, Ln 102-109, P. 3).
2. For the part of "Results", the framework of interviews had been shown. However, some key points should be summarised in each part. This can help the reader to easily understand the interview results.
Response: In accordance with your recommendations, we have summarised some key points at the end of each part of the ‘Results’ section. This now reads: “To sum up, the ‘pre-event’ phase articulates the false information regarding the safety issues of the Rana Plaza building that was spread among the workers by the factory management, which resulted in additional risks for individuals working during the day of the collapse.” (see ‘Results’, Ln 270-272, P. 11) “Thus, the ‘event’ epitomizes issues regarding the day of the collapse, including the influence of the factory management or building owner to encourage attendance at the workplace on the day of the collapse including the threat of no month-end salary with presence bonus and poor evacuation system existed inside the building.” (see ‘Results’, Ln 317-320, P. 12) Finally, the ‘post-event’ theme describes the ongoing physical and psychological health vulnerabilities, along with the overall wellbeing of the participants. It also focuses on the participants’ limited access to healthcare facilities and re-employment in the RMG sector, underlying which is the failure of the relevant agencies to ensure safety and security for those injured. ” (see ‘Results’, Ln 547-550, P. 17)
3. I think the authors can further compare the conditions of survivors before and after the accident. This may provide more information for potential implications of the current findings.
Response: The interview questions/guidelines were only related to the collapse of Rana Plaza which was mentioned in the ‘Materials and Methods’ section (Ln 127-133, P. 3). Therefore, this research did not focus on the survivors’ conditions before the accident.
4. For the part of "Discussion", it is no doubt that the survivors' physical and psychological health had been impaired after the accident. Therefore, the authors require to further explore whether new insights can be concluded from the findings. In addition, further implications for daily practices should be further elaborated.
Response: Following your suggestion, we have added a paragraph at the end of ‘Discussion’ section which now reads: “Signing the ‘Accord on Fire and Building Safety in Bangladesh’ on 15 May, 2013, by the international clothing brands and retailers as a consequence of the Rana Plaza collapse is probably the biggest achievement to result from the collapse to ensure the ongoing safety of the workers in the Bangladeshi RMG sector [5,6]. Through this Accord 80,000 hazards were identified in the RMG sector and 32 RMG factories closed after being found to be unsafe [7]. However, the continuation of this Accord, while it has the ability to increase safety, is in some doubt. The original timeframe for the Accord has ended (originally for five years ‘May 2013 to May 2018’ and later they were given an extension for six months as the interim period) [47] and without this international pressure, along with internal political issues, it is possible that factory workplace conditions will decline. Should this occur, another disaster, such as the one experienced at Rana Plaza, is inevitable. Thus, continued pressure to regulate this industry is required to ensure the safe work practices continue, and that those injured through such work are not discriminated against. This should be a shared concern among the community, the state, BMGEA, and all international brands commissioning work RMG work in Bangladesh. The findings of the study can be used to avoid the same kind of incident in a similar setup such as the South and Southeast Asian country perspectives, which house the majority RMG factories of the globe [23], and further protecting the daily activities of workers in this industry.” (see ‘Discussion’, Ln 602-617, PP. 18-19).
Reviewer 2 Report
Introduction
There needs to be additional justification added to the section to explain why this study is needed. This is an underrepresented population and region in the literature, thus this should also be stated.
Methods
Additional details regarding the qualitative methodology is warranted (e.g., where the interviews transcribed verbatim?)
What type of analytic lens did the authors use to identify and describe the themes listed in the result section?
Results
The authors should ensure that the length of this section is appropriate for the journal
Discussion
What are the implications of such reresearch for the broader sector and industry?
Author Response
Answer to reviewer
The authors are very grateful to the reviewer and to the editor for their suggestions to improve the quality of the article.
1. Introduction: There needs to be additional justification added to the section to explain why this study is needed. This is an underrepresented population and region in the literature, thus this should also be stated.
Response: Following your suggestion, we have added a justification of this study at the end of ‘Introduction’ section which reads: ‘Given the scope of the event and limited research on the topic, this study was conceived to better understand the impact on the health and wellbeing of the survivors. Findings from the study will provide insights to raise awareness of the workplace issues in the RMG industry, to help inform future prevention activities, and to ensure survivors needs are exposed to draw attention to adequate compensation and provision of appropriate healthcare services. As a result, the objectives of this study sought to identify the ongoing physical and psychological health vulnerabilities of the Rana Plaza survivors from their perspectives. It also examines access to, and uptake of, health treatment and compensation from both government and non-government organizations (see ‘Introduction’, Ln 102-109 P. 3)
2. Methods: Additional details regarding the qualitative methodology is warranted (e.g., where the interviews transcribed verbatim?). What type of analytic lens did the authors use to identify and describe the themes listed in the result section?
Response: Additional details are added to the ‘Materials and Methods’ section which now reads: ‘Interviews were tape-recorded and transcribed verbatim.’ (see ‘Materials and Methods’, Ln 113-114, P. 3).
‘The theme identification process, which used an interpretative lens, occurred in three rounds: the first review focused on reading and becoming familiar with the content, the second review was to identify themes, and the third review was to apply the theme categories once they were refined [32,33]’. (see ‘Materials and Methods’, Ln 173, P. 4).
3. Results: The authors should ensure that the length of this section is appropriate for the journal
Response: Thank you, this has been checked, and is within the author guidelines.
4. Discussion: What are the implications of such research for the broader sector and industry?
Response: Following your suggestion, we have added a paragraph on implications of such research at the end of ‘Discussion’ section which now reads: “Signing the ‘Accord on Fire and Building Safety in Bangladesh’ on 15 May, 2013, by the international clothing brands and retailers as a consequence of the Rana Plaza collapse is probably the biggest achievement to result from the collapse to ensure the ongoing safety of the workers in the Bangladeshi RMG sector [5,6]. Through this Accord 80,000 hazards were identified in the RMG sector and 32 RMG factories closed after being found to be unsafe [7]. However, the continuation of this Accord, while it has the ability to increase safety, is in some doubt. The original timeframe for the Accord has ended (originally for five years ‘May 2013 to May 2018’ and later they were given an extension for six months as the interim period) [47] and without this international pressure, along with internal political issues, it is possible that factory workplace conditions will decline. Should this occur, another disaster, such as the one experienced at Rana Plaza, is inevitable. Thus, continued pressure to regulate this industry is required to ensure the safe work practices continue, and that those injured through such work are not discriminated against. This should be a shared concern among the community, the state, BMGEA, and all international brands commissioning work RMG work in Bangladesh. The findings of the study can be used to avoid the same kind of incident in a similar setup such as the South and Southeast Asian country perspectives, which house the majority RMG factories of the globe [23], and further protecting the daily activities of workers in this industry.” (see ‘Discussion’, Ln 602-617, PP. 18-19).
Reviewer 3 Report
Some readers might be unfamiliar with Haddon's matrix. In the method section, Haddon's matrix could be explained briefly.
Author Response
Answer to reviewer
The authors are very grateful to the reviewer and to the editor for their suggestions to improve the quality of the article.
Reviewer 03:
Some readers might be unfamiliar with Haddon's matrix. In the method section, Haddon's matrix could be explained briefly.
Response: Following your suggestion, we have elaborated a bit more about Haddon’s Matrix, which now reads: “Following on from the development of the thematic schema, Haddon’s Matrix (a paradigm/theoretical framework best known in the injury prevention field to arrange events into a temporal sequence) [34] was utilized to further conceptualise the temporal range of the events in the participants’ narratives; that is, the themes were then organised into a pre-event, event, and post-event framework.” (see ‘Materials and Methods’, Ln 176-180, P. 4).
Round 2
Reviewer 1 Report
This revised manuscript shows an improvement when comparing with the previous one. However, few issues may need to be further addressed. My observation is shown as follows:
The findings state that follow-up medical services should be sufficiently provided. However, which kinds of medical services should be provided? The authors may need to further elaborate this point based on the current findings.
The findings also show potential reasons of this accident. However, the author should further explain why such reasons can lead to the accident, and practical suggestions can be provided to tackle such problems.
Hope that my comment is useful, and many thanks for giving me this opportunity to review this manuscript. Good luck.
Reviewer 2 Report
I am satisfied with the authors revisions and recommend the paper for publication
Author Response
The authors are very grateful to the reviewer.